# Effects of Living Grass Mulch on Soil Properties and Assessment of Soil Quality in Chinese Apple Orchards: A Meta-Analysis

Wenzheng Tang [1], Haosheng Yang [1], Wene Wang [1], Chunxia Wang [2], Yaoyue Pang [1], Dianyu Chen [1] and Xiaotao Hu [1,*]

[1] Key Laboratory of Agricultural Soil and Water Engineering in Arid and Semiarid Areas, Ministry of Education, Northwest A & F University, Yangling 712100, China; twh8811@163.com (W.T.); haoshengy@nwafu.edu.cn (H.Y.); wangwene@nwsuaf.edu.cn (W.W.); pangyaoyue@nwafu.edu.cn (Y.P.); dianyuchen@nwsuaf.edu.cn (D.C.)

[2] College of Water & Architecture Engineering, Shihezi University, Shihezi 832000, China; shdwchx@163.com

[*] Correspondence: huxiaotao11@nwsuaf.edu.cn

**Abstract:** Soil degradation has exacerbated the gap between crop yield and increasing food demands, and unreasonable field management is one of the main reasons for soil degradation. As a classic orchard soil management mode, living grass mulch can effectively change the hydrothermal environment and soil physicochemical properties of the 'soil–vegetation–atmosphere' microclimate of apple orchards. However, these improvement effects are mainly affected by climatic conditions, mulch methods, vegetation varieties and continuous grass-growing years. To evaluate the different effects of living grass mulch and the main influencing factors on soil physicochemical properties of apple orchards in China, in this study, we conducted a meta-analysis using data from 53 peer-reviewed publications to carry out soil quality assessment. The results showed that compared with clear tillage, continuous living grass mulch in apple orchards could improve soil function and performance by about 56% and increase soil enzyme activities by 10–120%, on average, whereas the soil organic matter under the effect of artificial grass and natural grass significantly increased by 29.6% and 14.6%, respectively. Artificial grass in temperate and warm, temperate, semi-humid climate regions had a greater overall improvement effect on the soil physicochemical environment than natural grass. Clover was found to be the most suitable for planting in apple orchards in temperate, semi-humid climate regions, whereas both clover and ryegrass were the best choices in warm, temperate, semi-humid climate regions. The interaction effects of different soil physicochemical properties in apple orchards in warm, temperate, semi-humid climate regions were greater than those in warm, temperate, arid climates and temperate, semi-humid climate regions. The response sensitivity of soil organic matter, organic carbon, urease, catalase, sucrose and cellulase to the living grass mulch effect of apple orchards was greater than that of other soil properties.

**Keywords:** apple orchards; living grass mulch; soil physicochemical properties; soil enzyme activities

## 1. Introduction

It is well known that apple cultivation occurs via asexual reproduction based on rootstocks. Choosing suitable rootstocks can ensure tree vigor and increase the full fruiting period. China is the world's largest apple producer [1–4], and its output increased sharply from $2.04 \times 10^5$ t in 2000 to $3.92 \times 10^5$ t in 2018 [5], which plays a crucial role in ensuring China's apple export and economic increase. However, in most apple-growing areas, the considerable economic benefits obtained by apple production come at the cost of gradual degradation of orchard soil quality [6]. Therefore, how to improve the soil environment and thereby enhance the soil quality is an urgent problem to be solved for the sustainable health development of China's apple industry.

Compared with mulch film, straw and other mulches, living grass mulch is more active and can regulate growth by cutting or crushing, which is an effective measure that improves soil physicochemical properties in orchards so as to enhance soil quality [7–9]. The practice of grass-growing management can be divided into whole-garden grass and inter-row grass [10], whereas the mulch method can be categorized as natural grass and artificial grass [11]. Since the introduction of orchard grass-growing technology in China in the 1990s [12,13], it has been widely applied in apple orchards, especially in the fruit-producing areas of southern China, where serious water soil erosion and soil impoverishment occur in red-yellow loam soil [14]. Natural grass is a convenient technology that uses the abundant grass species that grow naturally in orchards, with high grass yield and low cost. Artificial grass is a factitious selection of suitable and superior grass species for planting, which is conducive to manual control and management.

The choice of grass species considerably affects the improvement effects on soil quality [15,16]. At present, the common grass species in orchards are generally annual, biennial or perennial herbaceous plants [17,18], including clover, alfalfa, ryegrass, hairy vetch, fescue and crown vetch [19–21]. Different grass varieties have different functions.

A large number of studies [22–31] have confirmed that grass growing in orchards has many advantages, such as preventing soil erosion and hardening, increasing soil fertility, improving soil structure, increasing soil water-holding capacity, promoting soil biodiversity, and controlling weeds and pests. Therefore, grass-growing has considerable potential to improve soil quality. Although grass growing in orchards is also considered to have some shortcomings, such as competing with fruit trees for water and nutrients and time-consuming and laborious management [22,32,33], on the whole, it is an effective measure for orchard management. Except in extremely arid regions, grass growing has more advantages than disadvantages with respect to improving soil quality [34].

Field experiments are usually conducted independently in specific locations and therefore cannot be used to assess the comprehensive effects on a regional scale. Meta-analysis is a comprehensive statistical approach that can be used to synthesize independent test results and quantitatively evaluate the treatment effect on a regional or global scale [35]. In recent years, meta-analysis has been widely applied to the global or regional quantitative assessment of crop yield, soil water and nitrogen use efficiency, farmland soil greenhouse gas emissions and microbial functional diversity under specific treatment conditions [36–41]. However, there have been few reports on the comprehensive quantitative evaluation of the effects of living grass mulch in apple orchards on soil physicochemical properties in China at a climate region scale.

As a green measure with both environmental and economic benefits, grass growing is continuously increasing in popularity and being applied in apple orchards in China. However, due to the influence of factors such as climatic conditions, soil characteristics, fruit tree varieties and field management, the effects of grass growing often vary, with some cases of contradictory reports in the literature. Meta-analysis could represent a useful tool for the scientific evaluation of the influence of grass growing in apple orchards on soil physicochemical properties at a larger scale. Therefore, the objective of our study was to (1) quantify the effects of two main living grass mulch methods (artificial grass and natural grass) on soil physicochemical properties of apple orchards in China; (2) analyze the interaction effect of living grass mulch on different soil physicochemical properties in apple orchards; and (3) assess the soil quality.

## 2. Materials and Methods

### 2.1. Data Collection

We used "orchard (or fruit tree or apple) and grass-growing (or seeding or planting or inter-row or intercropping or mulching) and clean tillage" as the search keywords and the ISI-Web of Science (http://apps.webofknowledge.com/ (accessed on 15 March 2021)) and China Knowledge Infrastructure Project (https://www.cnki.net (accessed on 15 March 2021)) as the retrieval platforms. Peer-reviewed papers were searched to collect

relevant studies on the effects of grass growing in apple orchards on soil physicochemical properties in China until May 2020. The following criteria were used to select appropriate studies: (1) field-scale studies conducted only in China were included in this meta-analysis; (2) each study must include living grass mulch (treatment) and clean tillage (control) in apple orchards; and (3) Master's and doctoral dissertations without peer review were excluded. Based on these criteria, 53 publications (49 in Chinese and 4 in English) containing 705 side-by-side comparisons (643 for soil physicochemical properties and 62 for soil enzyme activities) were compiled into a dataset. Basic information of the selected peer-reviewed publications is listed in Supplementary Material.

Taking climate regionalization in China as the classification standard, in combination with China's diversified apple cultivation areas, we studied six types of climate regions: (1) temperate semi-arid; (2) temperate semi-humid; (3) warm temperate arid; (4) warm temperate semi-arid; (5) warm temperate semi-humid; and (6) subtropical humid. Table 1 shows the classification criteria for the six climate regions and corresponding basic soil and climate information.

**Table 1.** Classification standard of climatic regionalization in China and corresponding basic information of soil and climate in apple cultivation areas.

| Climate Region | | Temperate Semi-Arid | Temperate Semi-Humid | Warm Temperate Arid | Warm Temperate Semi-Arid | Warm Temperate Semi-Humid | Subtropical Humid |
|---|---|---|---|---|---|---|---|
| Classification index | ≥10 °C days * (°C) | 100–170 | 100–170 | 171–217 | 171–217 | 171–217 | 218–364 |
| | Annual dryness ** | ≥1.6, <3.5 | ≥1.0, <1.6 | ≥3.5, <16.0 | ≥1.6, <3.5 | ≥1.0, <1.6 | <1.0 |
| Climate information | Annual temperature (°C) | −4.2–8.0 | −4.2–8.0 | 8.0–13.0 | 8.0–13.0 | 8.0–13.0 | 13.0–20.0 |
| | Annual precipitation (mm) | 200–400 | 400–800 | 0–200 | 200–400 | 400–800 | ≥800 |
| Soil properties | AN (mg·kg$^{-1}$) | 7.0–191.0 | 12.2–114.6 | 8.1–92.3 | 41.5–93.0 | 22.7–183.7 | 56.1–245.1 |
| | AP (mg·kg$^{-1}$) | 2.5–95.8 | 14.5–44.5 | 21–89.3 | 3.5–15.9 | 5.0–221.0 | 1.7–185.1 |
| | AK (mg·kg$^{-1}$) | 84.3–315.0 | 19.3–242.8 | 58.0–497.4 | 65.4–171.3 | 71.8–248.1 | 50.8–229.0 |
| | SOM (g·kg$^{-1}$) | 1.6–20.5 | 15.1–83.6 | 2.1–32.2 | 0.6–30.0 | 6.4–13.8 | 4.7–59.2 |
| | SPH | 8.4–9.3 | 5.8–8.3 | 7.0–8.9 | 5.1–9.0 | 6.9–8.8 | 4.1–6.0 |
| | SBD (g·cm$^{-3}$) | 1.2–1.6 | 1.5–1.7 | 1.2–1.7 | 1.5–1.9 | 1.2–1.6 | 1.2–1.3 |

* represents the number of days that the sliding average temperature for many years will pass ≥10 °C; ** represents the dryness index, which refers to the ratio of maximum possible evaporation to precipitation; the maximum possible evaporation is calculated using the Penman (Penman, H.L.) formula [42]. SBD: soil bulk density; SPH: soil pH; SOM: soil organic matter; AN: soil available nitrogen; AP: soil available phosphorus; AK: soil available potassium.

Living grass mulch is applied in apple orchards in China under various different scenarios. Mulch methods can be divided into natural grass (natural growth) and artificial grass (artificial planting). Artificial grass includes 15 grass varieties (crown vetch, clover, ryegrass, alfalfa, fescue, dichondra repens, lotus corniculatus, dactylis glomerata L., hairy vetch, astragalus adsurgens, bermudagrass, bluegrass, chicory, bromus inermis and cocksfoot). To facilitate analysis and ensure a wide range of representativeness, the five grass varieties with the highest frequency (clover, alfalfa, ryegrass, fescue and crown vetch) were selected according to the research frequencies in our dataset. Continuous grass-growing years were divided into three categories: the initial stage (<3 years), mid-term (3–5 years) and long-term (>5 years). In addition, to make analysis results more reliable, classification groups with sample sizes of less than 4 were removed from the study.

### 2.2. Data Analysis

To quantitatively analyze the effect of living grass mulch in apple orchards on given variables, the natural logarithm (ln $R$) of the response ratio ($R$) was calculated as the effect size of this meta-analysis [35] and expressed as the grass-growing effect using the following equations:

$$R = \frac{X_t}{X_c} \tag{1}$$

$$\ln R = \ln \frac{X_t}{X_c} = \ln X_t - \ln X_c \tag{2}$$

where $X_t$ and $X_c$ are the mean of given variables in a living grass mulch treatment and corresponding clean tillage control, respectively. The average effect size and 95% bootstrap confidence interval (95% CI) were generated using Stata16.0. If CI did not overlap with zero, the treatment effect was considered positively (CI > 0) or negatively (CI < 0) significant [35]. Moreover, if CIs did not overlap among study categories, it was considered that there were significant differences in the treatment effect.

The standard deviations ($SD$s) of the given variables were not provided in most of the studies included in this meta-analysis. However, the interpolation of missing $SD$s has been proven to be accurate and safe [43,44]. The estimation procedure for the missing $SD$s involved calculating the $SD$ mean of given variables reported in included studies or similar research, which were expressed as corresponding ratios of the averages reported by specific variables; then, the value was multiplied by the reported average (missing $SD$) to obtain the appropriate $SD$.

The effect size variance based on $SD$ and sample size was calculated as:

$$V_{\ln R} = \frac{(SD_t)^2}{N_t(X_t)^2} + \frac{(SD_c)^2}{N_c(X_c)^2} \tag{3}$$

where $N_t$ and $SD_t$ represent the sample size and standard deviation, respectively, for living grass mulch treatment; $N_c$ and $SD_c$ represent the sample size and standard deviation, respectively, for clean tillage control; and $X_t$ and $X_c$ are the same as above.

According to the effect size variance, the standard error was calculated as follows:

$$SE_{\ln R} = \sqrt{\frac{V_{\ln R}}{N}} \tag{4}$$

where $N$ represents the sample size of each categorical variable group in the dataset, and $V_{\ln R}$ is the same as above.

Before analysis, it was necessary to perform a heterogeneity test on the dataset, which included two methods of heterogeneity statistics and homogeneity testing. In this study, we used heterogeneity statistics rather than homogeneity testing, mainly because the former can judge whether there was heterogeneity between studies and quantitatively analyze the degree of heterogeneity, whereas the latter can only qualitatively check whether there was heterogeneity between studies. Two heterogeneity metric indices for heterogeneity statistics based on a random effects model were calculated as follows:

$$I^2 = \frac{\tau^2}{\tau^2 + s^2} \times 100\% \tag{5}$$

$$H = \sqrt{\frac{\tau^2 + s^2}{s^2}} \tag{6}$$

where $\tau^2$ and $s^2$ represent the variance estimates within and between studies, respectively. According to the research of Higgins et al. [45], $I^2$ = 25% indicates low heterogeneity, $I^2$ = 50% indicates medium heterogeneity and $I^2$ = 75% indicates high heterogeneity. H < 1.2 indicates that there was no heterogeneity among studies, H > 1.5 indicates that

there was heterogeneity among studies and 1.2 < H < 1.5 when the 95% CI contains 1 indicates that the presence of heterogeneity could not be determined at the 0.05 level, whereas when it does not contain 1, heterogeneity is assumed [46]. In this study, the random effects model and DerSimonian–Laird estimation method were used for statistical analysis of heterogeneity, and the results showed that there was no heterogeneity (Table 2).

**Table 2.** Statistical results of variable heterogeneity.

| Variables | SBD | STP | SPH | SWC | SOM | SOC | TN | TP | TK | AN | AP | AK |
|---|---|---|---|---|---|---|---|---|---|---|---|---|
| $I^2$ (%) | 0.00 | 0.00 | 0.00 | 0.00 | 0.00 | 0.00 | 0.00 | 0.00 | 0.00 | 0.00 | 0.00 | 0.00 |
| H | 1.00 | 1.00 | 1.00 | 1.00 | 1.00 | 1.00 | 1.00 | 1.00 | 1.00 | 1.00 | 1.00 | 1.00 |

SBD: soil bulk density; STP: soil total porosity; SPH: soil pH; SWC: soil water content; SOM: soil organic matter; SOC: soil organic carbon; TN: soil total nitrogen; TP: soil total phosphorus; TK: soil total potassium; AN: soil available nitrogen; AP: soil available phosphorus; AK: soil available potassium.

Based on the basic theory and research method of Kuayakov et al. [47], the variation degree of soil characteristics was analyzed by adopting the soil quality index-area method (SQI-area) to evaluate the health status of the soil quality of apple orchards with living grass mulch. The method was calculated as follows:

$$stP_i = \frac{P_{clean\ tillage}}{P_{grassing}} \tag{7}$$

$$Area_{SQI} = 0.5 \cdot \sum_i^n stP_i^2 \cdot \sin\left(\frac{2 \cdot \pi}{n}\right) \tag{8}$$

where $stP_i$ is the standardized parameter, $i$ (i.e., each soil characteristic value should be standardized to 1.0 for apple orchards with living grass mulch); $P_{grassing}$ and $P_{clean\ tillage}$ are the mean of various soil characteristics under living grass mulch and clear tillage, respectively; $n$ is the total number of soil characteristics used for SQI and $\pi$ (3.14). SQI-area based on a radar map was suitable for the unified assessment and comparison of the variation degree of any number of soil characteristics. The ratio of the sum of all triangle areas in a radar map to its total area reflects the variation degree of soil quality.

## 3. Results

### 3.1. Overall Effects of Living Grass Mulch on Soil Physicochemical Properties of Apple Orchards in China

In China and compared with clear tillage, living grass mulch was found to significantly increase the STP (soil total porosity), SWC (soil water content), TP (soil total phosphorus), TK (soil total potassium), SOM (soil organic matter), SOC (soil organic carbon), AN (soil available nitrogen), AP (soil available phosphorus) and AK (soil available potassium) of apple orchard soils and significantly reduce the SBD (soil bulk density) and SPH (soil pH) (Figure 1). On average, SOM had the largest increase (25.99%), followed by SOC (25.96%), AN (16.46%), AP (16.08%), AK (13.91%), SWC (7.43%), TP (6.14%), STP (4.44%) and TK (1.80%). TN also increased by 6.08%, but the effects were not significant. SBD and SPH were significantly reduced by 5.25% and 0.46%, respectively.

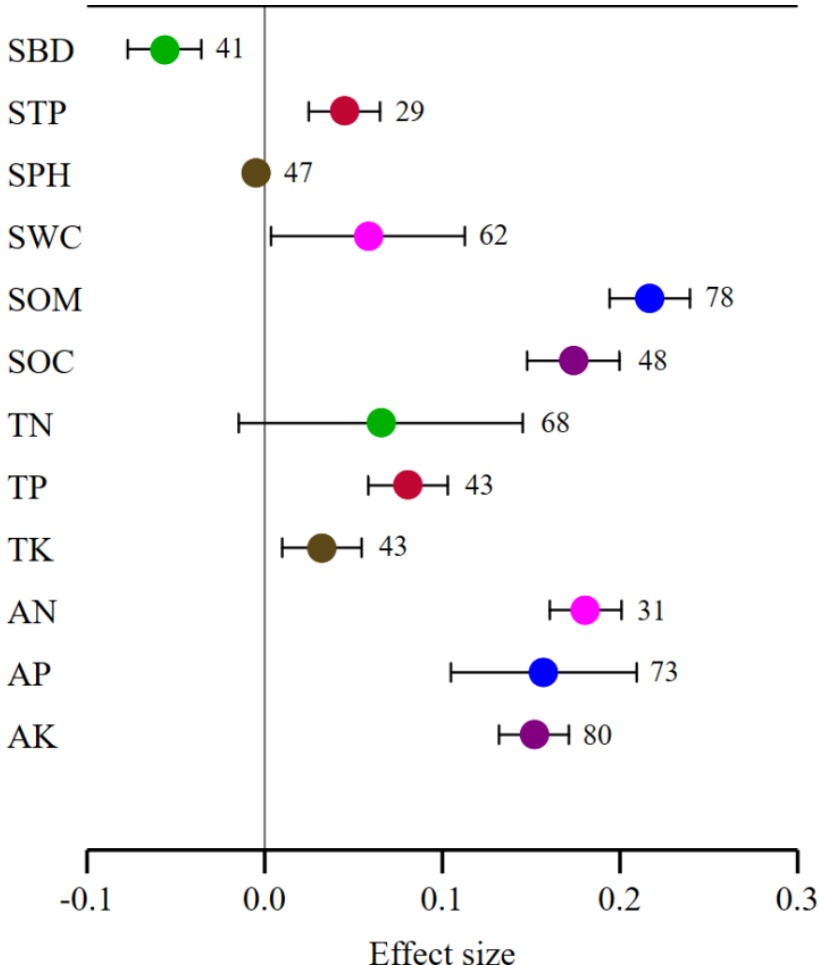

**Figure 1.** The overall effects of living grass mulch on soil physicochemical properties of apple orchards in China. Weighted means and their 95% confidence intervals of the effect sizes are given. The numbers on the right side of the confidence intervals represent the sample sizes. SBD: soil bulk density; STP: soil total porosity; SPH: soil pH; SWC: soil water content; SOM: soil organic matter; SOC: soil organic carbon; TN: soil total nitrogen; TP: soil total phosphorus; TK: soil total potassium; AN: soil available nitrogen; AP: soil available phosphorus; AK: soil available potassium.

### 3.2. Effects of Different Mulch Methods on Soil Physicochemical Properties of Apple Orchards in China

For apple orchards in China, the response of soil physicochemical properties to living grass mulch varied depending on the grass-growing method (Figure 2). Compared with clear tillage, both artificial grass and natural grass were found to significantly increase SOM, TP, AN, AP and AK. On average, the increase rates of SOM, TP and AK of artificial grass were higher than those of natural grass; in particular, SOM increased by 29.6%, which was twice that of natural grass (14.6%). Although the increase rates of AN and AP of artificial grass were lower than those of natural grass, the differences were not obvious. In addition, artificial grass significantly increased SWC and reduced SPH and non-significantly increased TN (soil total nitrogen) and TK. Natural grass increased TK and decreased TN significantly. Although it also increased SPH and SWC, the effects were not significant.

In addition to SPH, SWC, SOM, TN, TP, TK, AN, AP and AK, fewer than four data pairs were available for SBD, STP and SOC for apple orchards with natural grass in our dataset. Therefore, we only analyzed the changes in SBD, STP and SOC with respect to artificial grass. We found that artificial grass increased SOC and STP and decreased SBD significantly.

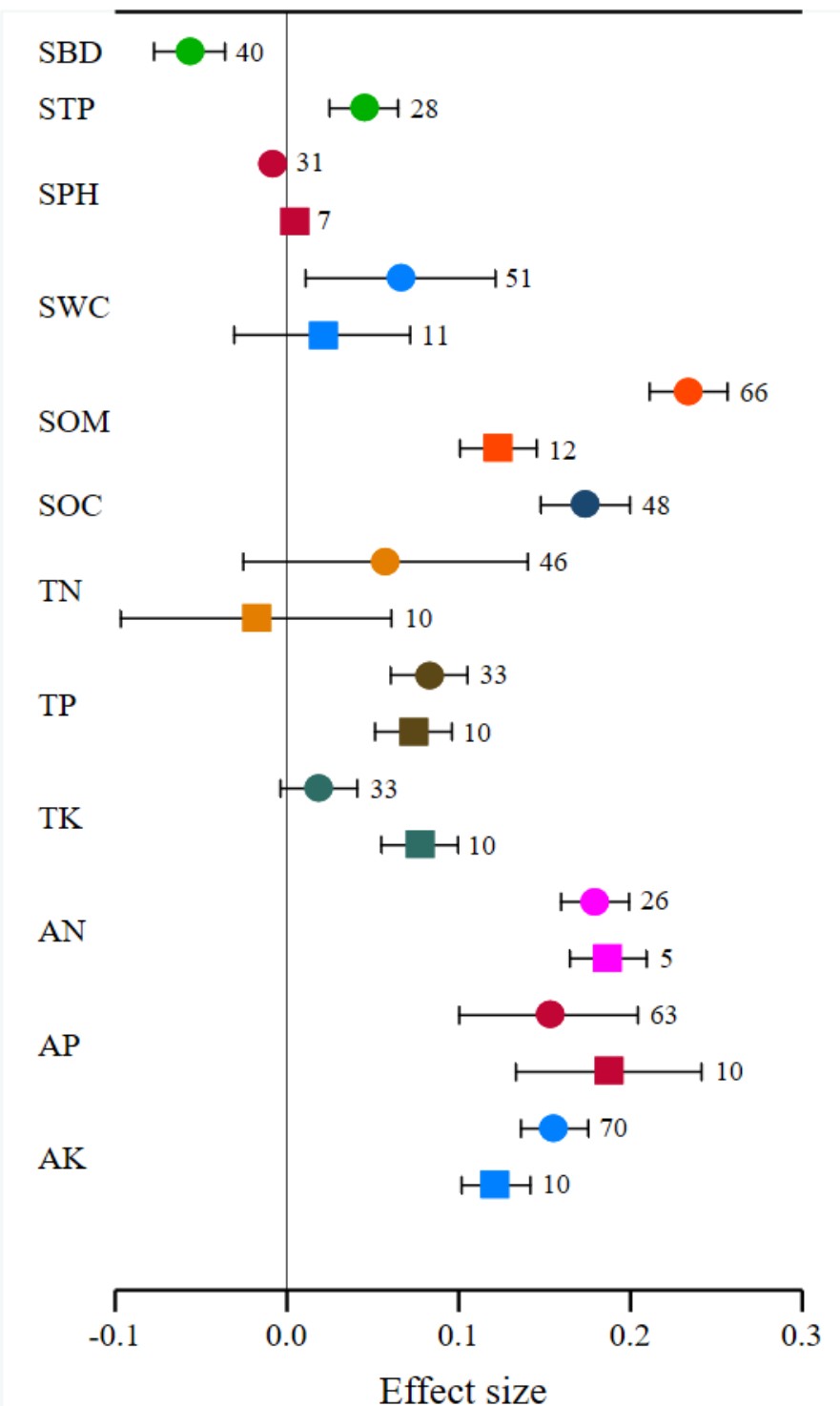

**Figure 2.** Effects of mulch methods on soil physicochemical properties of apple orchards in China. The circles and squares represent artificial grass and natural grass, respectively. Weighted means and their 95% confidence intervals of the effect sizes are given. The numbers on the right side of the confidence intervals represent the sample sizes. SBD: soil bulk density; STP: soil total porosity; SPH: soil pH; SWC: soil water content; SOM: soil organic matter; SOC: soil organic carbon; TN: soil total nitrogen; TP: soil total phosphorus; TK: soil total potassium; AN: soil available nitrogen; AP: soil available phosphorus; AK: soil available potassium.

### 3.3. Effects of Continuous Grass-Growing Years on Soil Physicochemical Properties of Apple Orchards in China

In China, the effects of living grass mulch in apple orchards on soil physicochemical properties vary with different continuous grass-growing years (Figure 3). In the initial stage of grass growing in apple orchards (<3 years), the SOM and AP increased significantly, and SWC also increased, although the effect was not significant. SPH, SOC, TN, and AK were basically stable, and the SBD decreased significantly. In the mid-term (3–5 years), SOM, SOC, AP and AK all increased significantly; TN also increased, although the effect was not significant; SPH was basically stable; SWC decreased to approximately the clean tillage level; and SBD decreased significantly. In the long-term (>5 years), SWC, SOM, SOC, TN, AP and AK all increased significantly; SPH increased slightly, although the effect was not significant; and SBD decreased significantly.

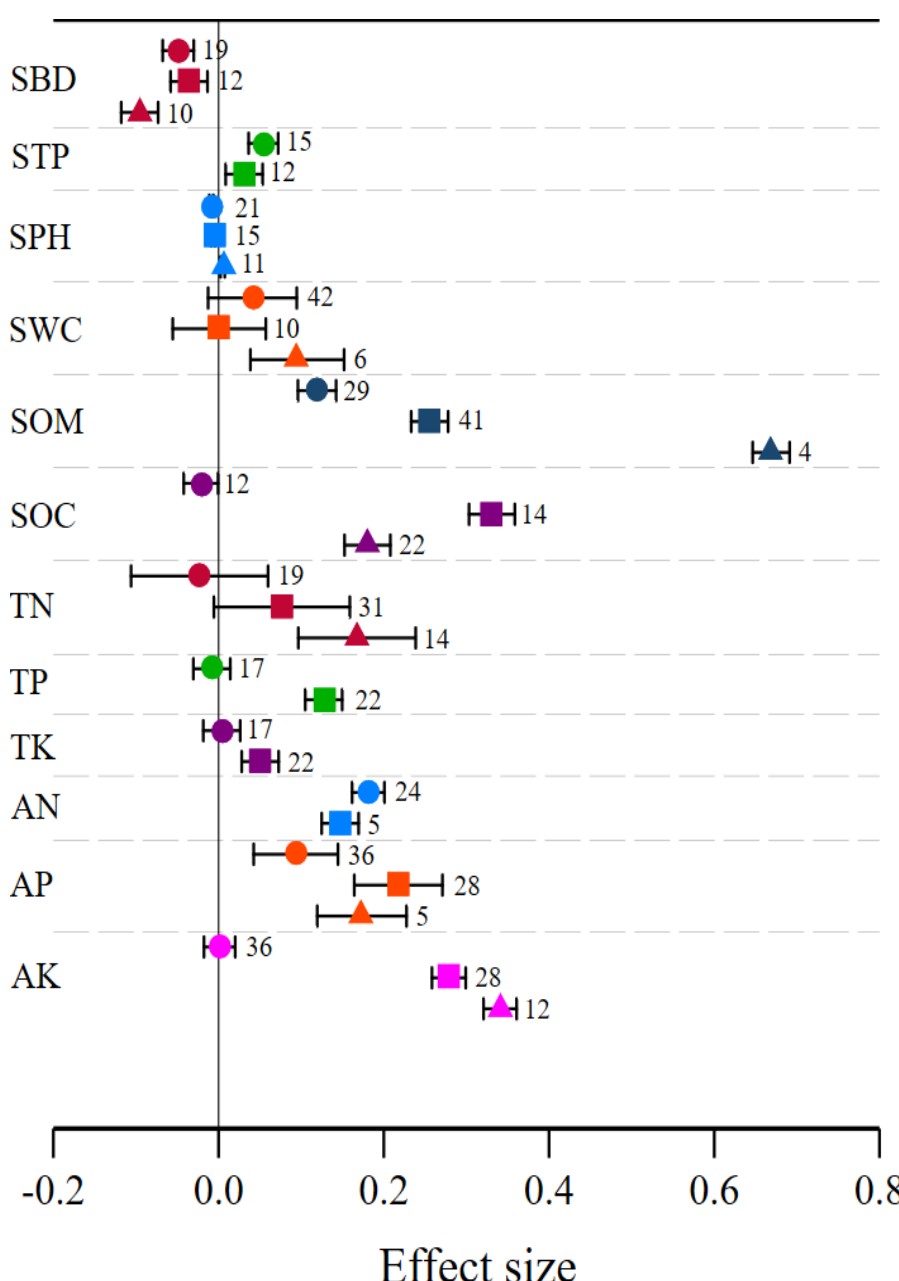

**Figure 3.** Effects of continuous grass-growing years on soil physicochemical properties of apple orchards in China. The circles represent the initial stage (<3 years), the squares represent the mid-term

(3–5 years) and the triangles represent the long-term (>5 years). Weighted means and their 95% confidence intervals of the effect sizes are given. The numbers on the right side of the confidence intervals represent the sample sizes. SBD: soil bulk density; STP: soil total porosity, SPH: soil pH; SWC: soil water content; SOM: soil organic matter; SOC: soil organic carbon; TN: soil total nitrogen; TP: soil total phosphorus; TK: soil total potassium; AN: soil available nitrogen; AP: soil available phosphorus; AK: soil available potassium.

Because there were fewer than four data pairs for STP, TP, TK and AN with respect to long-term grass growing in apple orchards in our dataset, we only analyzed their variation characteristics in the initial and middle stages of continuous grass growing. We found that both STP and AN increased significantly, whereas TP and TK were basically stable in the initial stage of grass growing (<3 years). In the mid-term (3–5 years), the STP, TP, TK and AN all increased significantly.

### 3.4. Response of Soil Physicochemical Properties to Different Influencing Factors for Apple Orchards with Living Grass Mulch in Climate Regions

In climate regions, besides the influence of local climate on soil physicochemical properties of apple orchards with grass growing, the effects of mulch methods, grass varieties and continuous grass-growing years could not be ignored. Studying the response of soil physicochemical properties in climate regions to different influencing factors will contribute to establishing an apple orchard grass-growing management system suitable for sustainable development under local climatic and soil conditions.

#### 3.4.1. Mulch Methods

Depending on the climatic region, the physicochemical properties of apple orchard soils respond differently to different mulch methods (Figure 4a). In a warm, temperate, semi-humid climate region and compared with clean tillage, SWC and AN were significantly increased in apple orchards with artificial grass, whereas natural grass only significantly increased AP. Whether artificial grass or natural grass, SPH was basically stable. Furthermore, the mean of STP, SOM, SOC, AP and AK with artificial grass increased significantly by 4.33%, 25.88%, 25.96%, 8.48% and 12.78%, respectively. In a temperate, semi-humid climate region and compared with clean tillage, both artificial grass and natural grass significantly increased SOM and TP in apple orchards, with average increases of 27.94% for SOM and 37.03% for TP under artificial grass, which is 3.3 and 4.1 times that of natural grass, respectively. In this climatic zone, TN and TK increased significantly under artificial grass, whereas they were basically stable and decreased under natural grass. Both artificial grass and natural grass increased SWC, although the increases were not significant, and the differences were not obvious. Moreover, SBD and SPH decreased significantly under artificial grass, whereas STP, AN, AP and AK increased significantly. In a warm, temperate, arid climate region and compared with clean tillage, SOM, AP and AK increased significantly under natural grass; TN decreased, although the effect was not significant; and TP decreased significantly. In a temperate, semi-arid climate region and compared with clean tillage, SOM and AK significantly increased under artificial grass, whereas AP increased, although the increase was not obvious.

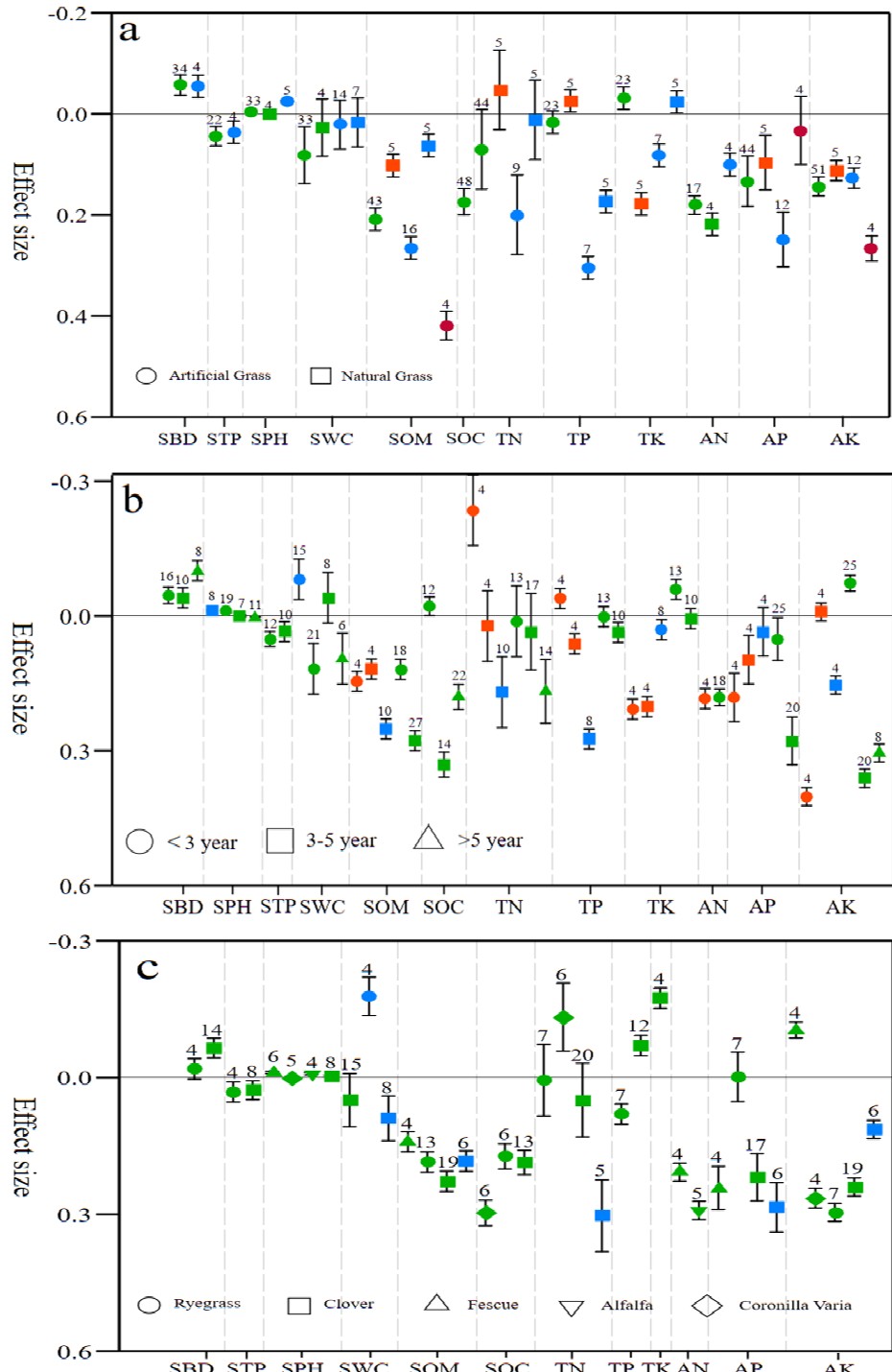

**Figure 4.** Response of soil physicochemical properties to different influencing factors for apple orchards with living grass mulch according to climate region. (**a**) Mulch methods. (**b**) Continuous grass-growing years. (**c**) Grass varieties. Medium green, orange-red, medium blue and cranberry colors represent warm, temperate semi-humid; warm, temperate arid; temperate, semi-humid; and temperate, semi-arid climate zones, respectively. Weighted means and their 95% confidence intervals of the effect sizes are given. The numbers at the top of the confidence intervals represent the sample sizes. SBD: soil bulk density; STP: soil total porosity, SPH: soil pH; SWC: soil water content; SOM: soil organic matter; SOC: soil organic carbon; TN: soil total nitrogen; TP: soil total phosphorus; TK: soil total potassium; AN: soil available nitrogen; AP: soil available phosphorus; AK: soil available potassium.

### 3.4.2. Continuous Grass-Growing Years

The effects of continuous grass-growing years on the physicochemical characteristics of apple orchard soils varied according to the climatic region (Figure 4b). In a warm, temperate, semi-humid climate region and compared with clean tillage, SWC, TP, SOM and AP all increased significantly in the initial stage of continuous grass growing (<3 years) in apple orchards; TN increased, although the effect was not significant; TP was basically maintained at the level of clean tillage; SOC decreased, although the decrease was not obvious; and TK and AK were reduced significantly. Compared with the initial stage (<3 years), STP, SOM and AP continued increasing significantly in the mid-term (3–5 years); and the SOC and AK, which were reduced in the initial stage (<3 years), also increased significantly in the mid-term (3–5 years). Similarly, TN and TP, which did not change obviously in the initial stage (<3 years), also increased significantly under the influence of mid-term grass planting (3–5 years). Compared with the initial stage (<3 years), SWC decreased slightly, although the decrease was not obvious; and TK gradually returned to the level of clean tillage. Compared with the mid-term (3–5 years), SWC, SOM, TN and AK were all increased significantly in the long-term (>5 years). Regardless of the number of continuous grass-growing years, SPH remained basically stable, and SBD was significantly reduced.

In warm, temperate, arid climate region and compared with clean tillage, TN and TP decreased in the initial stage (<3 years), whereas SOM, TK and AP increased significantly. Compared with the initial stage (<3 years), except for the continuedsignificant increase in SOM, TK and AP in the mid-term (3–5 years), TN and TP, which decreased during the initial stage (<3 years), were also increased under the effect of grass growing.

In a temperate, semi-humid climate region and compared with clear tillage, in the initial stage of grass growing in apple orchards (<3 years), SWC was reduced significantly, whereas TK and AK were increased significantly. In the mid-term (3–5 years), SOM, TN and TP were all increased significantly, and AP was also increased, although the average increase was not obvious. In this climate region, no variation tendency was observed for the same indices with respect to soil physicochemical properties between the initial stage (<3 years) and mid-term (3–5 years); therefore, it was not possible to determine the appropriate continuous grass−growth years for apple orchards in this climate region. However, in terms of changes in the existing soil physicochemical properties, during the initial stage (<3 years), SWC was reduced significantly, whereas in the mid-term (3–5 years), no unfavorable changes occurred in the soil SWC. In comparison, the mid-term (3–5 years) was found to be more suitable for continuous grass growing for apple orchards in this climate region. This might represent an appropriate means by which to determine the continuous grass-growing years in apple orchard cultivation areas in temperate, semi-humid climate regions.

### 3.4.3. Grass Varieties

The effects of grass varieties on soil physicochemical properties in apple orchards varied depending on the climatic region (Figure 4c). In warm, temperate, semi-humid climate regions and compared with clean tillage, planting fescue and ryegrass in apple orchards markedly increased SOM, although the average increase in SOM associated with planting ryegrass (32.54%) was much higher than that of fescue (14.53%). AK increased significantly when ryegrass was planted, whereas it decreased significantly under fescue. Planting ryegrass also significantly increased STP, SOC and TP. Planting crown vetch and ryegrass both significantly increased SOC in warm, temperate, semi-humid climate regions, whereas TN was reduced significantly under crown vetch and remained basically stable under ryegrass. Furthermore, planting ryegrass significantly increased STP, SOM, TP and AK, with average increases of 3.23%, 32.54%, 9.10% and 33.18%, respectively. The SOM and AP of apple orchards under fescue and clover were both increased markedly in warm, temperate, semi-humid climate regions, for which the average increases in AP were basically in agreement, whereas the average increase in SOM under clover (24.57%)

was much greater than that under fescue (14.53%). In addition, AK increased significantly under clover and decreased significantly under fescue. More importantly, planting clover increased STP, SWC, SOC and TN and significantly reduced SBD. Planting crown vetch and clover in apple orchards both significantly increased SOC in warm, temperate, semi-humid climate regions, whereas TN was decreased significantly under crown vetch and increased under planting clover. Moreover, planting clover increased STP, SWC, SOM, AP and AK and significantly reduced SBD. Planting clover and ryegrass in apple orchards in warm, temperate, semi-humid climate regions significantly increased STP, SOM, SOC and AK, with average increases of 3.23% and 3.26% for STP, 32.54% and 24.57% for SOM, 19.76% and 28.33% for SOC, and 33.18% and 25.61% for AK, respectively. Furthermore, SBD in apple orchards where ryegrass and clover were grown decreased by 1.92% and 5.86%, respectively, and the difference was not significant. However, in temperate, semi-humid climate regions and compared with clear tillage, SWC in apple orchards under ryegrass significantly decreased by 15.38%, whereas that under clover significantly increased by 9.81%. More importantly, planting clover also significantly increased SOM by 19.82%, TN by 20.84%, AP 30.06% and AK 7.75%.

### 3.5. Interactions among Soil Properties in Apple Orchards with Living Grass Mulch

In apple orchards with living grass mulch in China, SWC was significantly positively correlated with TN, whereas STP was significantly negatively correlated with SBD (Figure 5a). For apple orchards with living grass mulch in warm, temperate, semi-humid climate regions, SWC was significantly positively correlated with TN, TP, TK and AP. Similarly, STP was significantly positively correlated with SOC and TK, and there was a very significant positive correlation between SPH and AN and a very significant negative correlation with AK (Figure 5b). For apple orchards with living grass mulch in warm, temperate, arid climate regions, SOM and AK were extremely significantly positively correlated (Figure 5c). There was no significant correlation between soil physicochemical characteristics of apple orchards with living grass mulch in temperate, subhumid climate regions (Figure 5d).

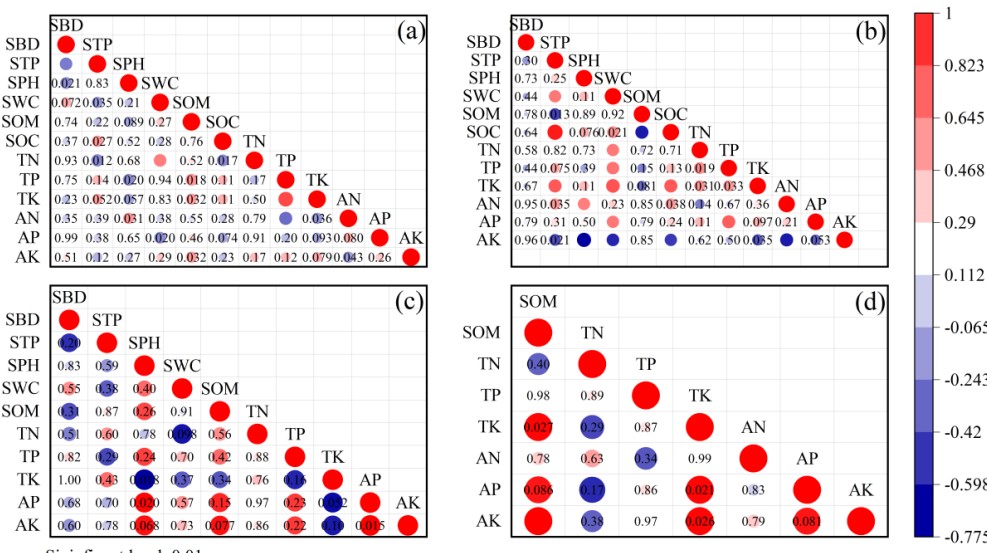

**Figure 5.** Correlation analysis among different soil characteristics. (**a**) the whole China region; (**b**) the warm temperature semid-humid climate region; (**c**) the temperature semi-humid climate region; (**d**) the warm temperature arid climate region. SBD: soil bulk density; STP: soil total porosity, SPH: soil pH; SWC: soil water content; SOM: soil organic matter; SOC: soil organic carbon; TN: soil total nitrogen; TP: soil total phosphorus; TK: soil total potassium; AN: soil available nitrogen; AP: soil available phosphorus; AK: soil available potassium.

### 3.6. Evaluation of Soil Quality in Apple Orchards with Living Grass Mulch

We found that if there was no continuous perennial grass growing in apple orchards, soil quality was severely degraded, with an average of about 56% (56%, Figure 6a; 55%, Figure 6b) of function and performance lost due to soil degradation. Similarly, both SOC and SOM increased by about 30%, and soil enzyme activities increased by 10–120% under the influence of continuous perennial grass growing (Figure 6). Furthermore, according to the reduction degree of different soil characteristics in clear tillage apple orchards compared with that under grass growing, the radar map indirectly reflects the most sensitive characteristics of soil quality in apple orchards to living grass mulch treatment. In this study, regardless of the research scale, the response sensitivity of SOM, SOC, SU (soil urease activity), SC (soil catalase activity), SS (soil sucrase activity) and SE (soil cellulase activity) to the living grass mulch effect of apple orchards was greater than that of other soil properties.

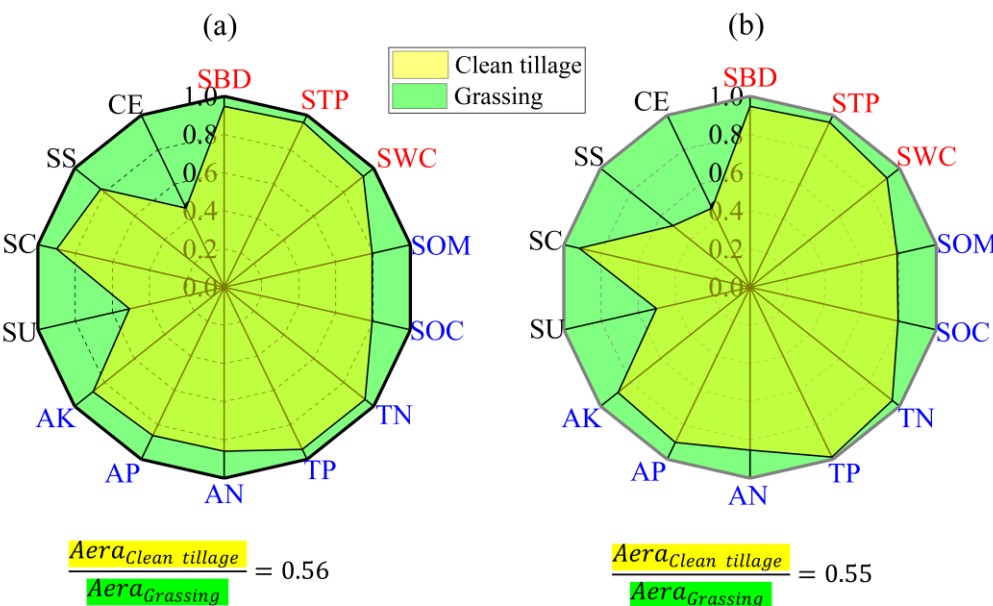

**Figure 6.** The soil quality index area (SQI-area) approach suitable for unifying evaluation of any number of soil parameters and to compare various SQI methods. (**a**) the whole China region; (**b**) the warm temperature semid-humid climate region. A decrease in the area on the radar plot between a non-degraded and a degraded soil is necessary. The ratio between the SQI area of non-degraded and degraded soils is independent of the number of parameters and the weightings involved in the calculation of the area. SBD: soil bulk density; STP: soil total porosity, SPH: soil pH; SWC: soil water content; SOM: soil organic matter; SOC: soil organic carbon; TN: soil total nitrogen; TP: soil total phosphorus; TK: soil total potassium; AN: soil available nitrogen; AP: soil available phosphorus; AK: soil available potassium; SU: soil urease activity; SC: soil catalase activity; SS: soil sucrase activity; SE: soil cellulase activity.

## 4. Discussion

### 4.1. Mulch Methods

Based on the analysis of data from 642 field experiments in China, it is clear that living grass mulch generally improves most soil physicochemical properties of apple orchards (Figure 1). SOM is the main source of soil nutrients, and change in its content comprehensively reflects soil quality [48,49]. Compared with clear tillage, living grass mulch improves SOM, as confirmed by experimental demonstrations, model studies and meta-analysis [50–52]. Our meta-analysis also confirmed this conclusion and highlighted that artificial grass can improve SOM in apple orchards more effectively than natural grass (Figure 2), which is consistent with the study result reported by Sánchez et al. [53] that natural grass had the lowest SOM compared with artificial grass. SOM mainly comes from

the input of surface litters and the decomposition of underground root systems [54–56]. Living grass mulch could contribute to the accumulation of SOM by greatly increasing the density and secretions of soil root systems in orchards [57,58]. Grass residues were incorporated into soil as an SOM supplement [59]. Living grass mulch has been found to help reduce soil erosion and increase SOC by slowing down the transportation and degradation of SOC during the precipitating process [9,60]. When grass wilts in autumn and winter each year, the carbon fixed in biomass of the grass is transferred to soil through a variety of physical and microbial processes [61]. Grass residues contribute considerably to the SOC pool by providing various SOM for soil [62,63], showing the overall positive effect of living grass mulch on SOC [64]. The network-type interspersed growth of grass roots in soil was found to be conducive to the formation of soil pores of different scales, which reduced SBD to a certain extent and increased STP [65], thereby improving the soil physical environment (Figures 2 and 4a). Grass stores rainwater by reducing surface runoff and increasing soil infiltration [66–68], thereby increasing SWC. Living grass mulch significantly increases AN, AP and AK in apple orchards (Figure 2), which might be related to nutrient storage balance between soil and plants. When the nutrient requirements of apple tree are basically met during whole growth period, the excess nutrients are stored in soil, and the high input of grass residues and nutrient transformation further improve soil nutrient reserves. In contrast to the increases of AN, AP and AK, the levels of TN, TP and TK all decreased. This might be due to the additional input of grass residues and conversion into total nutrients, leading to an increase; it is also possible that living grass mulch increased soil enzyme activities [69] and changed the soil microbial community [59,69] and utilization rate of microbial substrates [61,70], thereby promoting the conversion of total soil nutrients into available nutrients, leading to a reduction.

The effect of living grass mulch in apple orchards on the soil environment varies with according to the climatic region (Figure 4). These differences might be due to the synergistic results of the macroclimate in climate regions and the "soil–vegetation–atmosphere" microclimate in apple orchards. In both temperate and warm, temperate, semi-humid climate regions, the climates are cold and cool, the day–night temperature difference is large and light is sufficient, making them suitable areas for apple tree growth [71]. In temperate, semi-arid and warm, temperate, arid climate regions, apple tree irrigation is mainly rainfed, and drought and uneven precipitation seriously restrict the sustainable development of apple production in these regions. Microclimate could significantly reduce the surface evaporation of apple orchards and improve soil moisture retention capacity by decreasing surface temperature [72], wind speed and light transmittance [73], which are particularly important for the sustainable development of apple production in temperate, semi-arid and warm, temperate, arid climate regions. Although the growth and transpiration of grass inevitably consumes partial SWC, the research presented in the paper shows that compared with clean tillage, living grass mulch in apple orchards significantly increases SWC, which is basically consistent with previous research results [74,75], which fully proved that SWC consumption caused by grass growing is less than compensation. Microclimate could also regulate soil temperature of different soil layers in apple orchards [76,77], enhance the stability of soil temperature in the same soil layer and buffer abrupt changes in soil temperature [74], which not only promotes the growth of apple tree roots [78,79] but provides a favorable activity environment for soil micro-organisms and enzymes [80,81], which significantly affect the migration, absorption and transformation of various nutrients in the soil–plant system [82].

### 4.2. Continuous Grass-Growing Years

Continuous grass-growing years affect various soil physicochemical properties in apple orchards (Figure 3). With increased continuous grass-growing years, most soil physicochemical properties were significantly improved compared with those under clear tillage [83]; in particular, SOM and AP were significantly increased [84], which could be a comprehensive result of interactions between the rapid metabolism of grassroots, the continuous input

of residues, the activities of soil enzymes and microbial activities [15,85]. Compared with clear tillage, in the initial stage of grass growing in apple orchards (<3 years), SOM, AP and AN were found to be significantly increased, whereas TN, TP, TK and AK remained basically stable (Figure 3), which is not in agreement with existing research. In the existing literature, it has generally been reported that the initial stage of grass growing (<3 years) easily causes nutrient competition between fruit tree and grass, leading to soil nutrient loss. However, the revenue and expenditure balance between the amount of nutrients absorbed by grass and compensated by residues returning to soil should be taken into account in assessing changes in soil nutrients in apple orchards caused by grass growing. When the amount of nutrient uptake was greater than, equal to or less than the amount of nutrient compensation, soil nutrient changes were found to be decreased, stable and increased, respectively. In the mid-term (3–5 years), soil nutrients (TN, TP, TK and AK, etc.) increased significantly due to the positive bioenvironment created by the grassroot rhizosphere and the accumulation of residue and root exudates, which provided various nutrients for micro-organisms [86]. In the long-term (>5 years), the enrichment and decomposition of grassroots in root soil layers alternately occur in an annual cycle, which continuously improves the physical properties of SBD and STP in the root layers and indirectly or directly provides a favorable environment for improving soil nutrients and SWC. Similar results were found in citrus [87], pear [86], peach [88], apple [89] and apple orchards in our research. Furthermore, the interpenetration and expansion of grassroots to deep soil layers have been increasing year to year, which reflects that the impact of grass growing on the soil environment is a gradual process with an annual cycle. Only when continuous grass growing reaches a certain number years and the upper soil layer environment is significantly improved would this effect gradually advance to the deep soil layer.

Continuous grass-growing years have varying effects on the soil environment of apple orchards depending on the climate region (Figure 4b). Compared with clean tillage, TK and AK in warm, temperate, semi-humid climate regions and TN and TP in warm, temperate, arid climate regions were decreased significantly in the initial stage of grass growing in apple orchards (<3 years). This effect could be due to nutrient competition between grass and apple trees in the initial stage of grass growing (<3 years), leading to apple trees prestoring enough N, P and K to avoid nutrient shortage. Another possibility is that the suitable climatic conditions and microclimate effects in warm, temperate, semi-humid climate regions synergistically promote rates of use of N, P and K in apple trees. In addition, the SOC of apple orchards in warm, temperate, semi-humid climate regions was also found to decrease slightly, although the effect was not significant, which might be related to the additional carbon input, which could induce a priming effect to release $CO_2$ [90,91]. However, under continuous grass growing in apple orchards for more than 3 years, we found that most soil physicochemical properties in different climatic regions were significantly improved (Figure 4b), with living grass mulch gradually embarking on the path of sustainable development with a virtuous cycle.

### 4.3. Grass Varieties

The effect of grass on soil physicochemical properties varies according to the climate region [92,93]. Our study showed that the improvement effect of soil physicochemical properties associated with clover planting in apple orchards in temperate, semi-humid climate regions was better than that of ryegrass, whereas both clover and ryegrass were better than crown vetch, alfalfa and fescue in warm, temperate, semi-humid climate regions (Figure 4c). This could be attributed to the differences in the growth habits and climate-adaptable capabilities of different grass species. Ryegrass has both poor properties with respect to cold and heat resistance, whereas temperate, semi-humid climate regions have high temperatures, with rainy summers and severely cold winters with little snow, which is not conducive to ryegrass growth. On the contrary, clover is more resistant to cold and drought and prefers humid and warm climates [94]. Furthermore, clover is a leguminous,

perennial grass, and its roots accumulate plenty of rhizobia to fix nitrogen in air, enhance its nitrogen-fixing capacity and, as green manure, eventually significantly increase the AN of apple orchards. Exudates produced by root systems could also increase SOM to a certain extent (Figure 4c). Warm, temperate and temperate, semi-humid climate regions have similar climatic characteristics, although the former is relatively warmer and can adapt well to the growth habits of both ryegrass and clover, which can both significantly improve most soil physicochemical properties in apple orchards (SBD, STP, SOM, SOC and AK). With the current background of global warming, warm, temperate, semi-humid climate regions experience frequent strong rainfall in summer, with orchard soils easily forming surface ponds over a long period of time. Both crown vetch and alfalfa are intolerant to waterlogging, and surface ponding for 3–4 days can cause their roots to rot and die, which is unfavorable for growth. The whole growth period of fescue and apple basically coincide, causing simultaneous competition for water and nutrients, eventually leading to a reduction in apple yield [22].

### 4.4. Correlation Analysis among Soil Characteristics in Apple Orchards of Living Grass Mulch

The reasons for changes in various soil physicochemical properties in apple orchards based on the effect of living grass mulch were analyzed and explained above (Sections 4.1–4.3). However, physicochemical properties did not change independently of the effect of living grass mulch. When one or more of soil physicochemical properties changed, others were induced to change accordingly, with the soil physicochemical influencing and interacting with each other (Figure 5). An increase in SWC was found to be beneficial to decomposition and nutrient release of grass residues and roots. In Chinese apple orchards, SWC was very significantly positively correlated with TN, whereas in warm, temperate, semi-humid climate regions, it also showed very significant positive correlation with TN, TP, TK and AP (Figure 5). This could be the result of the combined effect of research scale and climate differences. Apple cultivation areas in China are widely distributed across a variety of climate types. Climate type and soil texture have varying degrees of influence on the decomposition rate of grass residue and the conversion rate of different soil nutrients. In contrast, the distribution of apple cultivation areas in warm, temperate, semi-humid climate regions is relative concentrated and affected by a single climate type with a relatively consistent degree of influence. In addition, the STP under grass growing in apple orchards in China was significantly negatively correlated with SBD and positively correlated with SOC in warm, temperate, semi-humid climate regions, which is similar to the research results of Rosa et al. [95] and Deurer et al. [96]. In apple orchards under living grass mulch in warm, temperate, semi-humid climate regions, SPH and AN were found to have be very significantly positively correlated (Figure 5b), mainly due to the suitable hydrothermal environment in the region, which increased soil microbial activities [97], promoted decomposition of grass residue and released compounds $H^+$ and $Al_3^+$ [98], resulting in a tendency for alkaline SPH, which inhibited transformation of $NH_4^+$ and $NO_3^-$ in the soil, indirectly improving AN. However, SPH and AK were significantly negatively correlated, which is consistent with the research results of Xie et al. [99].

### 4.5. Soil Quality Assessment of Apple Orchards with Living Grass Mulch

Soil quality is a combination of physicochemical and biological properties in soil and some key processes related to the formation of these properties [100]. Our study showed that living grass mulch in apple orchards significantly improved soil quality, and the response sensitivity of SOM, SOC and soil enzyme activities (SU, SS, SC and SE) to the effect of living grass mulch was much greater than that of other soil properties (Figure 6), which was verified in some related studies [101–103]. Soil enzymes are a kind of bioactive substance with catalytic ability, which are released into soil as a result of the decomposition of animal and plant residues, as well as micro-organisms and their secretions [104,105]. They are regulators of soil biochemical processes and play an important role in the formation of SOM [59,106]. SU affects soil nitrogen mineralization by promoting

urea hydrolysis [107,108]. SS is related to contents of SOM, nitrogen and phosphorus in soil; microbial amount; and soil respiration intensity, and its enzymatic products are directly related to plant growth. SC promotes the decomposition of hydroperoxide in soil to relieve its toxic effects on soil organisms, thereby providing a suitable soil environment for plant root growth and soil microbial activities. SE plays an important role in the soil carbon cycle [109].

## 5. Conclusions

Based on the summary of relevant studies reported in China, further analysis showed that compared with clean tillage, living grass mulch can significantly improve the soil environment of apple orchards. The improvement effect of artificial grass on different soil physicochemical properties in apple orchards was found to be better than that of natural grass, independent of climate type. In warm, temperate, arid or sub-humid climate regions, Continuous grass growing for more than five years could significantly improve most soil physicochemical properties in apple orchards, and SOM and SOC were also found to increase significantly. In terms of living grass mulch in apple orchards in temperate, semi-humid climate regions, clover was found to be the best grass species, whereas in warm, temperate, semi-humid climate regions, clover and ryegrass were found to be the best choices. Under the effect of living grass mulch, the interaction effect among different soil physicochemical properties in temperate, semi-humid climate regions was much greater than that in warm, temperate, arid or temperate, semi-humid climate regions. Living grass mulch in apple orchards could significantly mitigate soil quality degradation. The response sensitivity of SOM, SOC, SU, SC, SS and SE to living grass mulch was found to be greater than that of other soil characteristics. In apple field systems in China, grass mulching is an effective strategy for sustainable production of apples and improves soil quality according to the local soil environment, climatic conditions and management methods.

**Supplementary Materials:** The following supporting information can be downloaded at: https://www.mdpi.com/article/10.3390/agronomy12081974/s1.

**Author Contributions:** Conceptualization, X.H. and W.W.; methodology, W.T. and D.C.; validation, H.Y. and X.H.; formal analysis, W.T., C.W. and Y.P.; investigation, W.T. and H.Y.; writing—review and editing, W.T., D.C. and X.H.; supervision, X.H. and W.W.; funding acquisition, X.H. and W.W. All authors have read and agreed to the published version of the manuscript.

**Funding:** This research was funded by the National Natural Science Foundation of China grant number [51179163] and Natural Science Basic Research Program of Shaanxi Province of China grant number [2022JZ-24] and The APC was funded by [51179163] and [2022JZ-24].

**Institutional Review Board Statement:** Not applicable.

**Informed Consent Statement:** Informed consent was obtained from all subjects involved in the study.

**Data Availability Statement:** The authors declare that the data supporting this study are available from the corresponding author upon reasonable request.

**Acknowledgments:** We are thankful to all the researchers whose contributions were used in our study analysis and referenced in this review article.

**Conflicts of Interest:** The authors declare no conflict of interest.

## Abbreviations

SBD: soil bulk density; STP: soil total porosity, SPH: soil pH; SWC: soil water content; SOM: soil organic matter; SOC: soil organic carbon; TN: soil total nitrogen; TP: soil total phosphorus; TK: soil total potassium; AN: soil available nitrogen; AP: soil available phosphorus; AK: soil available potassium; SU: soil urease activity; SC: soil catalase activity; SS: soil sucrase activity; SE: soil cellulase activity.

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
