# Peer review of "Effects of Living Grass Mulch on Soil Properties and Assessment of Soil Quality in Chinese Apple Orchards: A Meta-Analysis"

_agronomy, doi:10.3390/agronomy12081974_

Round 1
Reviewer 1 Report
About the introduction: few information about rootstocks should be included. How is the percentage of clonal rootstocks (including dwarfing type) compared to seedlings rootstocks in China; living mulch can be very dynamic and that the growth can be controlled with mowing or crushing; soil fertility include soil sickness problems.
About the results and discussion it should be taken in to account that apple suffers for replant problems and so soil quality should include soil sickness. The same approach is for type of living mulch. The same species alone cannot stay forever in the same soil. Never you produce clover indefinitely, in open field farmers make rotation to maintain a high yield. Why this problems should be not taken into account using clover or another single species as living mulch in apple orchards? Living mulch rotation or mixing of several species in the orchard. so the positive effect is high in the first years and than does not increase in the long term.
Information about rodents should be included.
English must be revised in a more clear version of some phrases and spelling
Other literature to be consulted
Effect of soil condition on apple root development and plant resilience in intensive orchards. Polverigiani S., M. Franzina, D. Neri, 2018. Applied Soil Ecology, 123: 787–792 DOI:10.1016/j.apsoil.2017.04.009. In APPLIED SOIL ECOLOGY - ISSN:0929-1393 WOS:000437772500066
Growth of ‘M9’ apple root in five Central Europe replanted soils. Polverigiani S., Kelderer M., Neri
Soil nitrogen and weed biodiversity: An assessment under two orchard floor management practices in NVZ (Italy). Md Jebu Mia, Elga Monaci, Giorgio Murri, Francesca Massetani, Jacopo Facchi, and Davide Neri (2020). Horticulturae, 6, 96; doi:10.3390/horticulturae6040096
Reviewer 2 Report
The aim of the work is to assess the differences in the physicochemical properties of soil
on apple plantations in China between living grass mulch and clear tillage.
The study uses data from other studies, published mainly in Chinese.
The data from the work were divided into climatic regions,
taking into account the species of plants in the apple row spacing,
whether the grass was of natural or artificial origin.
The work is valuable, in particular because it presents results that a non-Chinese-speaking reader cannot access.
It is written legibly, but there are linguistic errors,
e.g. "represent the dryness index, which refer to the ratio of maximum possible
evaporation to precipitation". I suggest that it be checked by a native speaker.
"the maximum possible evaporation is calculated using the Penman (H. L., Penman) formula" reference is needed.
The authors base the results and conclusions on the confidence intervals (CI) for
the difference of the logarithm of the variables determining the physicochemical properties of soils with and without grass.
They are only CI comparisons, if they contain zero then there is no grass effect,
if they are negative the effect is negative (variable discreases), and if positive it increases.
In my opinion, there are so many possible better statistical analyzes for this type of comparison,
e.g. multifactorial analysis of variance.
The second thing is that such designated CIs give the reader little information.
Why are the mean and standard deviations determined by the authors not presented in the paper?
They are collected in the supplementary materials, but they are in no way statistically compiled.
I suggest changing the methods of data analysis or justifying why this method is better for assessing research hypotheses.
Round 2
Reviewer 2 Report
The authors explained my doubts.